# Development of Novel Experimental Models to Study Flavoproteome Alterations in Human Neuromuscular Diseases: The Effect of Rf Therapy

**DOI:** 10.3390/ijms21155310

**Published:** 2020-07-26

**Authors:** Maria Tolomeo, Alessia Nisco, Piero Leone, Maria Barile

**Affiliations:** Department of Biosciences, Biotechnology and Biopharmaceutics, University of Bari, 70126 Bari, Italy; maria.tolomeo@uniba.it (M.T.); alessia.nisco@gmail.com (A.N.); pieroleone87@gmail.com (P.L.)

**Keywords:** *SLC52As*, *FLAD1*, mitochondrial flavoproteome, flavoprotein subunit-SDH, riboflavin, RTD, MADD, LSMFLAD, RREI, model organisms

## Abstract

Inborn errors of Riboflavin (Rf) transport and metabolism have been recently related to severe human neuromuscular disorders, as resulting in profound alteration of human flavoproteome and, therefore, of cellular bioenergetics. This explains why the interest in studying the “flavin world”, a topic which has not been intensively investigated before, has increased much over the last few years. This also prompts basic questions concerning how Rf transporters and FAD (flavin adenine dinucleotide) -forming enzymes work in humans, and how they can create a coordinated network ensuring the maintenance of intracellular flavoproteome. The concept of a coordinated cellular “flavin network”, introduced long ago studying humans suffering for Multiple Acyl-CoA Dehydrogenase Deficiency (MADD), has been, later on, addressed in model organisms and more recently in cell models. In the frame of the underlying relevance of a correct supply of Rf in humans and of a better understanding of the molecular rationale of Rf therapy in patients, this review wants to deal with theories and existing experimental models in the aim to potentiate possible therapeutic interventions in Rf-related neuromuscular diseases.

## 1. The Human Flavoproteome

Riboflavin (Rf, also known as vitamin B2), a water-soluble vitamin belonging to the B-group vitamins, is the precursor of flavin cofactors FMN (flavin mononucleotide) and FAD (flavin adenine dinucleotide), which allow for cellular flavoproteome to become enzymatically active.

The molecule of Rf (7,8-dimethyl-10-ribitylisoalloxazine) consists of a substituted isoalloxazine ring, whose N-10 atom is bound to a ribityl residue. In FMN the 5′ end of the ribityl moiety is esterified by a single phosphoryl group; adenylation of FMN gives rise to FAD. The chemical structures of Rf and of its derived cofactors are reported in Figure 1, in the context of their transport and metabolism.

The human flavoproteome is constituted by the products of more than 90 genes, namely hundreds of different flavoenzymes, most of which (84%) utilizing, as a cofactor FAD, while only 16% depend on FMN. There are some less represented cases in which both cofactors are concomitantly required by the same enzyme, as, for example, in the case of methionine synthase reductase [1].

The role of flavin cofactors in the enzymatic catalysis lies essentially in transferring reduction equivalents, giving rise to the semi-reduced or fully-reduced forms, thus FMN and FAD are essential molecular constituents of a large number of dehydrogenases, reductases and oxidases, mainly located in mitochondria, being involved in intermediary and terminal energetic metabolism of fatty acids, carbohydrates, amino acids, pyridoxine and choline [2]. Nevertheless, certain non-redox flavin dependent reactions have been recently identified, as the isomerization of UDP-galactopyranose to UDP-galactofuranose by UDP-galactopyranose mutase or the reaction catalyzed by the peroxisomal alkyl-dihydroxyacetone phosphate synthase involved in the biosynthesis of ether phospholipids [3].

The ability of flavin enzymes to catalyze a wide-range of redox reactions in a variety of biological processes relies on the electrochemical tuning of the enzyme-bound flavin redox cofactor. Enzyme active sites are able to modulate the flavin redox potential generally from +100 mV to −400 mV, spanning a 500 mV range that enables flavoenzymes to catalyze a variety of redox reactions [4].

The versatility of flavin-dependent oxidoreductases explains why flavoenzymes play a key role in redox homeostasis in both generating and scavenging reactive oxygen species (ROS) and reactive nitrogen species (RNS) [5]. Thus, on the one hand, we can cite the complex of NADPH oxidase, which becomes functionally active in the plasma membrane of activated macrophages [6]. On the other hand, we can remember the flavoenzymes involved in the recycling of glutathione [2,7], as well as thioredoxin reductase [8].

In the majority of cases, apo-flavoenzymes bind flavin cofactors non-covalently (90%), whereas in some cases, a covalent linkage occurs. Interestingly all the covalent flavinylated enzymes are confined inside organelles, first of all mitochondria, with succinate dehydrogenase (SDH, respiratory chain complex II, EC 1.3.5.1), dimethylglycine dehydrogenase, sarcosine dehydrogenase and monoamine oxidase being the most extensively investigated (see [9,10,11,12] and refs therein).

In the intermembrane space of mitochondria, the FAD-dependent apoptosis-inducing factor (AIF) triggers caspase-independent programmed cell death [13] and also, a FAD-dependent pathway controls oxidative protein folding [14].

An even more intensively investigated FAD-dependent pathway controls protein folding in the endoplasmic reticulum (ER) [15], under the control of FAD trafficking [16]. In the nucleus, FAD-dependent oxidases play some roles in chromatin remodeling and epigenetic events, controlling the expression of genes involved in energy metabolism [17,18].

Moreover, Rf has a central role in pyridoxine metabolism [19], in folate and Vitamin B12 recycling [20,21,22] and therefore in one-carbon metabolisms, as well as in biosynthesis and regulation of coenzyme A, coenzyme Q_10_, heme, steroids and thyroxine [23].

## 2. Rf Absorption and Cell Delivery

Since humans cannot synthesize Rf, its dietary supplementation is essential [7]. Rf is present in many foods such as green vegetables, dairy products, eggs and meat. Nevertheless, a certain amount of Rf can even be supplied endogenously by some microbes residing in the large intestine. Its recommended daily intake is 1.3 mg per day for men and 1.1 mg per day for women with some variations depending on age and physical state (e.g., pregnancy or lactation). Rf status can be altered by malabsorption occurring in different conditions, such as celiac disease, malignancies, and alcoholism.

Rf absorption takes place mainly in the small intestine and partly in the large intestine [24] and it occurs via specific carrier-mediated processes, supported by three members of the solute carrier family 52A. They are named Rf transporter 1 (RFVT1), RFVT2 and RFVT3, which perform different functional and kinetical properties [25].

*SLC52A1* was the first gene identified coding for a human Rf translocator [26]: it is located on chromosome 17 at 17p13.2. Soon after *SLC52A2* and *SLC52A3* were also cloned and characterized as reported in [27,28]. These two genes are located one on chromosome 8 at 8q24.3 and the other on chromosome 20 at 20p13, respectively.

In enterocyte, which allows for a trans-epithelial vectorial Rf movement to portal circulation, the vitamin is taken up by the action of RFVT3 at the apical membrane and it is released in blood by RFVT1 and RFVT2 located at the basolateral membrane [29]. Before crossing the intestinal barrier and entering both the circulating and the peripheral cells, Rf is converted into FMN and FAD via the sequential action of riboflavin kinase (RFK, EC 2.7.1.26) and FAD synthase (FADS, EC 2.7.7.2) (see [30,31] and below).

It should be noted that RFVTs mediate the translocation of Rf, rather than FMN or FAD [32]. FMN and FAD, derived from digested proteins, must be converted into Rf again before being delivered into enterocytes. This task is performed by extracellular diphospho- (EC 3.6.1._) and monophospho- hydrolases (EC 3.1.3.2), located on the intestinal brush border. These hydrolytic events occur later on, before Rf transport into specialized cell of peripheral tissues [2,19]. A comprehensive study on the identity of these “transport-preparing” hydrolases is missing at the moment. Quite unspecific activities working on the plasma membrane surface were initially proposed to perform this function by [33]. The discovery of the big family of Nudix hydrolases (Nudix stands for *“*nucleoside diphosphate linked to some other moiety X hydrolase”), as well as that of a FAD-diphosphatase (FADDPase) activity “hidden” in the N-terminus domain of the bi-functional FAD synthase [34,35], suggests the possibility for the existence of a flavin-linked specific process; this necessitates further research.

Through the portal venous system, Rf reaches the liver and is taken up by hepatocytes, possibly using a Na^+^-independent and energy-dependent carrier regulated by intracellular Ca^2+^/calmodulin as showed in HepG2 cell line [36], as well as in pancreatic β cells/islets [37]. However, the contribution of each RFVT in the human hepatic homeostasis of Rf has not yet been fully clarified, apart from data reported in “The human protein Atlas database” suggesting the prevalence of RFVT2. Similar conclusions have been also reported in the mouse model [38].

Circulating Rf is bound both to plasma albumin (*k*_d_ = 3.8–10.4 mM) and more tightly to a subfraction of immunoglobulins [39] with median plasma concentrations of 10.5, 6.6 and 74 nmol/L for Rf, FMN and FAD [40].

Excess of vitamin intakes or of tissue requirements is excreted in the urine as Rf or other metabolites [41]. Due to the regulation of absorption and excretion, the circulating Rf level is subjected to a small circadian variation [42].

## 3. Rf Transporters

### 3.1. Some Molecular Insights on RFVTs

The recently identified RFVTs have different tissue-specific expression profiles as well as functional and kinetical properties [25]. Investigations on the expression profile of RFVTs at the transcriptional level revealed that RFVT1 is particularly abundant in the placenta and small intestine, RFVT2 is ubiquitously expressed, but it is relevant especially for the brain, while RFVT3 is expressed mostly in intestine and testis [28].

In addition, we would like to speculate that alternative splicing of each gene, might generate different products, some of which are not yet described at the protein level. Currently, in Entrez Gene database (www.ncbi.nlm.nih.gov/gene) three RefSeq transcripts are reported for *SLC52A1*, all encoding for the same 448 aa protein (Figure 2). Eleven transcript variants are reported for *SLC52A2*, among which one is a non-coding RNA, six encode for the same 445 aa protein, three encode for the same putative 281 aa protein and one encodes for a 357 aa protein (Figure 2). According to this database, a single protein of 469 aa is encoded, starting from four different transcript variants generated by *SLC52A3* (Figure 2). Recently, two *SLC52A3* transcript variants that differ in the transcriptional start site, were described in esophageal squamous cell carcinoma (ESCC), named SLC52A3a and SLC52A3b. The first variant corresponds to NM_ 033409.4 in the RefSeq database. The second variant encodes for a protein of 415 aa (Figure 2). Immunofluorescence analysis of SLC52A3a and SLC52A3b in ESCC cell lines revealed that SLC52A3a is localized in the cell membrane and in the nucleus, while SLC52A3b is found in the cell cytoplasm. A stronger distribution of SLC52A3a in the nucleus has been correlated with poor prognosis in ESCC patients [43].

All three RFVTs have been predicted to have 11 trans-membrane (TM) domains. Further details are reported in [2,44,45]. A novel homology model of hRFVT1 is presented here (Figure 2). The *N*-glycosylation status of RFVT3 has been addressed in [46]. Mutating the two predicted *N*-glycosylation sites at Asn^94^ and Asn^168^, and the substrate-interacting residues identified by the protein-docking modeling approach, lead to impairment in Rf uptake and to intracellular (ER) retention of the mutated proteins in HuTu-80 cells, demonstrating the importance of glycosylation and structure-function relationship in physiology/biology of intestinal epithelial cells.

As far as transport mechanisms and kinetic features of Rf uptake are concerned, the first series of studies were carried out with different natural cells. For an exhaustive review on these studies, the reader is referred to [47]. After genes’ identification, some additional features of transport have been clarified, using cells transfected with RFVTs’ cDNA. The uptake of Rf by all RFVTs was proven to be independent on extracellular Na^+^ and Cl^−^ [28], in accordance with previous reports [47], but in contrast with reports in [48]. The pH sensitivity was also tested, indicating that the sole RFVT3-mediated uptake is pH-dependent [27,28].

As for RFVTs’ inhibitors: Rf transport activity is inhibited by flavin analogs, such as the natural cofactors FMN and FAD and the artificial molecule lumiflavin (7,8,10-Trimethylisoalloxazine) in HEK-293 cells [26]. Lumichrome (7,8-Dimethylalloxazine) and amiloride also inhibited Rf transport in NCM460 cells [49]. In transfected HEK-293 cells, hRFVT-mediated uptake of Rf resulted in complete inhibition by an excess of Rf and lumiflavin but modest inhibition by FMN. Whereas, only hRFVT2- mediated Rf uptake resulted slightly but significantly inhibited by FAD [28]. In addition, RFVT3-mediated Rf uptake was also inhibited by amiloride, ethidium, and methylene blue, but not by d-ribose and alloxazine [50]. Besides all findings obtained with intact cells, the recombinant RFVT2 and the native protein extracted from fibroblasts and reconstituted in proteoliposomes are also inhibited by FMN and lumiflavin [44]. This is the first example of studies with a purified recombinant human plasma membrane vitamin translocator.

Transcriptional mechanisms regulating Rf transport in different organs and tissues are not completely elucidated, being the intestine epithelial cells the first and the best model, because of the relevance of vitamin absorption in human nutrition. A big piece of work was made in this sense by the Said group (see below).

The first approach to address this problem was aimed to identify the minimal promoter regions of *SLC52A1* and *SLC52A3*, the main Rf intestinal transporters, together with the regulatory elements involved in their activation. Analysis performed in the promoter region of *SLC52A1* showed that its core activity is embedded in the region between −234 and −23 bp and it contains several putative cis-regulatory sites, including KLFs, AP-2, EGRF, and stimulating protein-1 (Sp-1). A significant decrease in promoter activity was found mutating each of the cis-regulatory sites of *SLC52A1* promoter, with a more pronounced reduction for Sp-1. Focusing on the Sp-1 site, Electrophoretic mobility shift assay (EMSA), super-shift and Chromatin immunoprecipitation (ChIP) analysis performed on HuTu-80 cells and studies with *Drosophila* SL-2 cells confirmed the important role of Sp-1 in regulating the activity of *SLC52A1* promoter [51].

The same analyses performed on *SLC52A3* revealed the core promoter activity encoded between −199 and +8 bp, and it includes binding sites for NF-κB/cRel, KLF, and Sp-1. Among these putative cis-regulatory elements, only Sp-1 was found to play an important role in HuTu-80 cells. Studies with *Drosophila* SL-2 cells showed that besides Sp-1, Sp-3 can also be led to *SLC52A3* activation. Moreover, with the use of luciferase gene fusions, the activity of the cloned *SLC52A3* promoter was confirmed in vivo in transgenic mice [52]. Differently from the bulk of the existing knowledge about the regulatory regions of *SLC52A3* and *SLC52A1*, responsive elements in the *SLC52A2* promoter are still not characterized. Furthermore, with the use of the Caco-2 and HuTu-80 cells, the Said group demonstrated that RFVT3 is a target for post-transcriptional regulation by miR-423-5p, which interacts with the 3′-UTR region of hRFVT3, leading to a decrease in the translational efficiency and in intestinal Rf uptake. Similar results were obtained with mouse intestinal enteroids [53].

### 3.2. Rf Transporters: What Else

Using human intestinal epithelial NCM460 cells grown in Rf-depleted and over-supplemented media, Said group also demonstrated that the intestinal Rf uptake process can be regulated by extracellular substrate level. Rf-deficient conditions resulted in an increased expression of hRFVT-2 and -3 (but not hRFVT-1) and of Sp-1 at both the protein and the mRNA levels. ChIP assay for histone H3 modification revealed a significant decrease in the activity of the heterochromatin marker (H3K27me3) in cells maintained in Rf-deficient conditions, suggesting the possible involvement of epigenetic changes in the *SLC52A3* promoter [54].

Recently, using Caco-2 cells and mouse colonoids, sodium butyrate (NaB) has been demonstrated to up-regulate intestinal Rf uptake inducing the expression of *SLC52A3*, possibly due to epigenetics alteration in *SLC52A3* promoter after NaB treatment [55]. Besides its important role as an energetic substrate in intestinal epithelial cells, butyrate is involved in the regulation of immune cell by modulating different processes of intestinal epithelial cells and leukocytes through the activation of G protein-coupled receptors and the modulation of the activity of enzymes and transcription factors including the histone deacetylase [56]. An additional mode of action of NaB could involve the increase of expression of TMEM237, a transmembrane protein encoded by the *ALS2CR4* gene, which directly interacts with RFVT3 ensuring its stability [57].

Another piece of work concerning regulation of RFVTs expression level has been made in relationships to modulation by TNF-α, one among the best-characterized activators upstream of NF-kB signaling pathway. TNF-α addition to Caco-2 cells reduced the expression of RFVT1 and 3 [58]. Its effect could be also mediated via a reduction of the level of expression of the TMEM237 protein [57]. The relevance of NF-*k*B in regulating RFVT3 expression, upon TNF-α stimulation due to Rf deficiency has been recently proposed also in ESCC [43]. Altered expression patterns of *SLC52As* in human colorectal cancer, involving different transcriptional and post-transcriptional mechanisms specific for each transporter, have been reported ([59] and refs. therein).

A further membrane transporter responsible for Rf movement across the plasma membrane is the less specific ABCG2 transporter, which mediates the secretion of Rf in milk and other extracellular fluids [60]. The ABCG2 transporter belongs to the family of ABC translocators (ATP-binding cassette G2 transporter) and it is also called BCRP (breast cancer resistance protein), since it was first isolated from a multidrug-resistant breast cancer cell line. Its specific tissue distribution is closely linked to the physiological role it assumes as, for instance, limiting absorption (in the intestine), mediating distribution (in the blood-brain and blood-placental barriers) and facilitating elimination and excretion (in the liver and kidney) of a wide variety of drugs, molecules carcinogenic and food toxins [61]. The dual action of BCRP both as a detoxifier and as an Rf transporter has been preserved in the course of evolution, so that it can be found from lower mammals to humans [62].

A main question, still completely under-investigated, concerns the subcellular trafficking of Rf. The existence of a FAD recycling pathway, which implies a carrier-mediated process for Rf across the inner mitochondrial membrane, was demonstrated by our group in mammals, yeasts and plants [63,64], but the nature of the mitochondrial Rf translocator is still unknown.

## 4. Rf Intracellular Homeostasis

### 4.1. Biochemical Pathways of FAD Synthesis and Degradation

Enzymes metabolizing Rf to its derived cofactors i.e., Rf kinase (RFK) and FAD synthase or FMN:ATP adenylyl transferase (FADS or FMNAT) in humans are encoded by two distinct genes, named as reported in Figure 1.

*RFK* gene, located on chromosome 9 at 9q21.13, codes for a 17.6 kDa polypeptide consisting of a single domain, whose kinetics is largely regulated by the relative concentration of substrates/products [65]. Its crystal structures (PDB codes: 1NB0, 1NB9, 1P4M, 1Q9S), confirmed that the nucleotide-binding motif [30] undergoes large conformational changes upon binding of Rf, that is to say the reaction is highly regulated by the vitamin taken up [30,66]. These data, together with observations made in RFK-KD (knock down) cells [18], allowed us to propose that RFK is the limiting step of the intracellular conversion of riboflavin into FAD [2]. This flavin-based kinetic control of the flavin cofactor forming enzyme could be exerted together with other regulatory mechanisms, such as those triggered by thyroid hormones [67].

RFK deficiency has never been described in humans, therefore it was suggested that it might be lethal, as previously reported in mice [68]. However, hypomorphic *RFK* mutations might also result in a clinical presentation similar to that seen in patients with FADS deficiency [23]. Indeed, conditional RFK knockout strains of mice have been generated to address the impact of acute Rf deficiency on host defence against *Listeria monocytogenes* [69]. Moreover, knockdown of RFK combined with a Rf deficient diet has been proven to alter the levels of cryptochrome protein in mouse liver and the expression profiles of the clock and clock-controlled genes. Therefore, light-independent mechanisms depending on FAD contribute to the proper circadian oscillation of metabolic genes in mammals [70].

*FLAD1* gene, located on chromosome 1 at 1q21.3, codes differently sized polypeptides, consisting of either a single or two domains, corresponding to different transcript variants generated by alternative splicing of the gene. When the human gene was initially identified by our group [31,71], two protein isoforms (namely hFADS1 and hFADS2) were described. hFADS1, encoded by the seven exons long transcript variant 1 (GenBank Accession n: NM_025207.5), is a 587 aa protein with a predicted molecular mass of 65.3 kDa; the first 17 residues of this isoform represent a putative mitochondrial-targeting peptide, as predicted by bioinformatic analysis [72]; hFADS2 is a 490 aa protein with a predicted molecular mass of 54.2 kDa, which lacks a 97-mer in the N-terminal region of hFADS1 with a cytosolic localization. hFADS2 is the product of transcript variant 2 (GenBank Accession n: NM_201398.3), the most abundant in tissues/cells tested so far, which derives from interruption of exon 1 by an additional intron [72] (Figure 3).

Both hFADS1 and 2 contain an N-terminal molybdopterin binding (MPTb, recently renamed FADHy [35] domain, since it hides a FAD hydrolase activity [34,35], which is fused with a C-terminal 3-phosphoadenosine 5-phosphosulfate reductase domain (PAPS, recently renamed FADSy [35]), which *per se* performs the FADS activity [73,74,75].

Isoform 2 has been over-produced and purified in its catalytically active form: it contains redox-sensitive cysteines which make hFADS2 a putative redox-sensor [76]. In [35], the novel feature of hFADS2 to catalyze FAD hydrolysis has been characterized especially for its ability to create a link between the flavin and the NAD world as already proposed in yeast [7,77]

Besides regulating FAD production/hydrolysis, FADS also takes part in cofactor delivery to the appropriate apo-flavoenzymes during holoenzyme biogenesis, operating in a flavinylation machinery as a FAD “chaperone” [78,79].

Of course, the two opposite processes, i.e., FAD synthesis and hydrolysis, both performed by the same polypeptide (hFADS2), can constitute a “futile cycle”, thus they cannot work contemporarily; it would be very intriguing to discover the molecular mechanisms controlling the switch of synthesis versus hydrolysis of FAD, that is to say, the changing in the relative concentration of FAD vs. FMN in vivo.

### 4.2. The Puzzle of the Splicing Variants of FLAD1 and the Sub-Cellular Origin of FAD

The number of the transcript variants of *FLAD1* is far to be defined. Besides transcript variants 1 and 2, two additional RefSeq transcript variants are reported in the Entrez Gene database, as schematized in Figure 3. The transcript variant 3 (GenBank accession n: NM_00114891.2) differs in 3′ end compared to transcript variant 2 resulting in a 446 aa protein with a shorter C-terminus. It is very similar to the isoform 2, therefore it is presumably able to perform both FAD synthesis and FAD hydrolysis.

The transcript variant 4 (GenBank accession n: NM_001184892.2) has multiple differences and initiates translation at an alternative start codon, compared to variant 2. The protein product of 294 aa (isoform 4) is shorter and has distinct N- and C-termini when it is compared to hFADS2.

An additional protein (isoform 5), reported exclusively in UniprotKB, is a 338 aa protein and it is very similar to isoform 4. The last two isoforms, containing the sole FADHy domain, are expected to have hydrolytic activity, but these isoforms, as well as the protein corresponding to isoform 3, have not been either produced or characterized yet.

Moreover other three transcript variants (ENST00000295530.6, ENST00000368428.1, ENST00000368433.5) are annotated in ENSEMBL genome browser 100 (https://ww.ensembl.org/index.html).

The existence of additional “orphan” isoforms was indeed postulated since 2013 [7] and further confirmed in 2016 [80] when two novel transcript variants, not yet annotated, were revealed by transcriptomic analysis. They correspond to the sole C-terminus protein domain and were named isoform 5 and 6 [80] (Figure 3). Isoform 6 was produced as a recombinant protein in its FADSy active form and its enzymatic action was characterized in some detail, confirming the absence of a hydrolase activity in this novel [73].

The contemporary expression of different *FLAD1* products in the same cells [7] was interpreted as a mode to regulate the sub-cellular compartmentation of the process of FAD delivery to apo-flavoproteins [7,78], but further studies are necessary to strengthen and confirm our observation/proposal.

Besides the already stressed mitochondrial localization of hFADS1, in a number of different mammalian cells, a nuclear localization for FAD forming enzymes has been demonstrated [81] and correlated to the emerging role of FAD and derived cofactors as regulators of epigenetic redox events [82]. In the nucleus, FAD hydrolytic events were also described, but we do not yet know if they are due to one of the bi-functional isoforms generated by *FLAD1* or if they occur via independent hydrolases [81]. Unfortunately, the identification of the nuclear FADS isoform is still lacking, as well as its possible alteration in Rf-related neuromuscular disorders. An intensive effort, therefore, urges to better understand these potentially crucial nuclear events.

Moreover, some intracellular hydrolytic activities i.e., FADDPase and FMN hydrolase are localized in the inter-membrane space of mitochondria. They were characterized at the functional level, but not identified at the molecular one; they accompany Rf re-cycling in rat and human muscle mitochondria [63,83,84] via a still-unidentified mitochondrial Rf transporter. Thus, the process of flavin transport across mitochondrial membrane is a matter of debate.

In humans, FAD transport across the mitochondrial membrane requires a specific inner membrane carrier encoded by *SLC25A32*, which is located at chromosome 8 at 8q22.3. Three RefSeq transcript variants are reported for this gene, only one encoding for a protein of 315 aa.

It has been initially identified as a mitochondrial folate transporter [85], and then as the human orthologue of the yeast mitochondrial FAD transporter FLX1 [86,87] which allow cytosolically synthesized FAD to enter mitochondrial membrane and *vice versa* [88].

Mutations in this gene are causative of a neuromuscular disorder responsive to Rf treatment [89,90], whose phenotype is described below.

The role of *SLC25A32* in the field of cancer research has recently emerged [91].

## 5. Remaining Challenges in Neuronal and Muscular Flavin Homeostasis and their Alterations

### 5.1. Rf Neuronal Homeostasis and BVVLS

Given the importance of vitamin B2 in oxidative metabolism crucial for nervous system economy, the mechanisms of Rf transport and homeostasis in the nervous system have been long investigated in the past (see [92] and refs therein) and further clarified on the discovery of the human hRFVT2, which immediately appeared to be maximally expressed in the human brain and spinal cord [28,93].

Utilization of glucose, certain amino acid and ketone bodies’ metabolism, as well as aminoacidic neuromediator synthesis are crucial for brain (Figure 4). Utilization of fatty acids by the brain has been, for a long time, excluded based on the impermeability of the blood-brain barrier. More recently, the importance of brain economy of short-chain fatty acid, as butyrate emerged and, thus we can imagine that flavin-dependent β-oxidation is the key pathway involved in butyrate-utilizing cells [94].

Thus, Rf deficiency is expected to slow-down all these processes, thus nervous tissue alterations/degeneration are expected as biochemical consequences of FAD forming deficiencies, as depicted by [95]. Nevertheless, given the cellular heterogeneity of the brain, it is not surprising that different cell types have a distinctive Rf demand.

Rf enters the brain from blood at the blood-brain barrier (BBB) via a saturable system (*k*_m_ ~0.1 μM) and this process presumably involves RFVT2 [48,96], but, as far as we know, a direct demonstration of the sole presence of this transporter is lacking. Moreover, other translocators (OAT3 and ABCG2) which control Rf efflux from the Choroid Plexus seem to be additional candidates in ensuring a constant flavin supply in the extracellular space of the brain [60,61,92]. In CSF (cerebrospinal fluid), total flavin concentration is reported to be ~0.1 μM, with the sole Rf concentration being ~0.02 μM [96,97]. A systematic study concerning alterations of these values in both control individuals and in patients suffering from Rf deficiency is still missing. The possibility that ancillary cells might be involved in Rf delivery to neurons has to be considered. In fact, the vast majority of cells in the adult human brain are astrocytes, which are intimately associated with synapses and govern synapse formation and plasticity [98].

The importance of Rf supply for neuronal cells clearly emerged in 2010, when the alteration in *SLC52A3* was correlated to Brown-Vialetto-Van Laere Syndrome (BVVLS) [99]. BVVLS is an early-onset disease characterized by progressive cranial neurons loss (resembling amyotrophic lateral sclerosis (ALS)), degeneration of spinal cord neurons and respiratory insufficiencies. This disease, now named Riboflavin Transporter Deficiency 3 (RTD3, OMIM #211530), is considered to be the same as the Fazio-Londe disease (OMIM #211500) [100], which differs from BVVLS only for the lack of hearing loss. Not long after *SLC52A3* disease gene discovery, mutations in *SLC52A2* have also been associated with BVVLS, now named RTD2 (OMIM #614707) [101,102].

Other common clinical features present in RTD2 and RTD3 patients are dysarthria, weakness and hypotonia, whereas the most common differences concern facial weakness which is typical of RTD3 patients and vision loss, characteristic of RTD2 patients [103,104]. RTDs are sometimes responsive to high doses of Rf treatment and characterized by biochemical abnormalities in the acylcarnitine profiles, thus resembling another inborn error of metabolism mainly affecting muscle, named MADD (Multiple Acyl-CoA Dehydrogenase Deficiency) we will discuss later on [105].

Since RTD definition, a total of 109 RTD patients have been recently exhaustively reviewed in [103]; further nine novel cases have been reported in [106,107,108,109,110,111,112,113,114].

A clear relationship between Rf intracellular scarcity and biochemical damages in neurons is still lacking, as well as the molecular mechanism of regulation of the flavin transport efficiency in a different area of the brain and in different types of neurons. Another unknown issue is the sub-cellular localization of both the flavin transporters and the enzymes involved in homeostasis maintenance.

First investigations, at the cellular level, to address these points concern (i) mimicking RTD deficiencies in HEK-293 transfected cells [101] and (ii) evaluating transport defects in fibroblasts from patients [93]. For a complete summary of this kind of study, the reader is referred to [103].

Since mitochondrial dysfunction has widely been implicated in several neurodegenerative disorders [115,116], a piece of work has been carried out to link the effects of Rf transport loss of function with mitochondrial function and neuronal integrity. This was achieved by in vitro study performed on BVVL patients’ fibroblasts, muscular biopsy and induced pluripotent stem cells (iPSCs) differentiated in the motor neuron.

A significant reduction in the intracellular levels of FMN and FAD has been observed in RTD2 patient fibroblasts when grown in low extracellular Rf conditions, together with impaired electron transport chain complex I and complex II activities. [117]. Consistently, in muscle biopsies of both types of RTD patients, mitochondrial respiratory chain deficiencies were revealed [118,119,120]. All these data are consistent also with a mitochondrial myopathy, which was confirmed by muscle histopathology revealing ragged-red fibers.

The question of whether RFVT3 can be directly involved in the neuronal and muscular supply of Rf or rather a secondary intestinal absorption deficiency causes the myopathy, is still an open question [105].

To better investigate the neuronal functional and structural consequences, RTD patient-specific iPSC-derived motoneurons have been developed. A reduction in axonal elongation, partially improved by Rf treatment, and a perturbation in neurofilament composition have been observed accompanied by a reduced autophagic/mitophagic flux [121].

Further evidence of mitochondrial dysfunction have been described in the *Drosophila melanogaster* model (see [117] and below).

Turning back to neuronal flavin homeostasis, quite surprisingly almost no data exists about neuronal RFK and FADS. The presence of the *FLAD1* transcript isoform 1, as well as the enrichment in these cells of a protein of about 65 kDa (as estimated by SDS-PAGE) [122] allow us to propose that mitochondrial neuronal FAD synthase exists, as well as other isoforms, whose subcellular localizations are still obscure. Moreover, confocal microscopy experiments previously performed by our group, provided clear evidence of FADS expression in rat neonatal astrocytes, with a cytosolic and a strong nuclear localization [81].

We would like to remind that, BVVLS is defined as a juvenile form of ALS (OMIM #105400), a fatal degenerative disease affecting upper motor neurons of the cortex and lower motor neurons of the brainstem and spinal cord. Quite interestingly, a small but significant decrease in mRNA encoding FADS, together with RFK and some proteins of the electron transport chain, was observed in the blood of patients suffering from ALS in comparison with healthy people [123]. In the same paper, a 65 kDa FADS was recognized as the target antigen in an ALS patient who had at the same time a monoclonal gammopathy. The putative surface membrane localization of FADS on motor neurons [123] seemed in contrast with the canonical intracellular localization [72,81].

Even if these results do not allow understanding the link among FADS expression or FAD isoforms and the disease, it is plausible that alterations of specific neuronal FADS isoforms could cause a sub-optimal energy metabolism that could make the patient’s motoneurons more susceptible to degeneration.

Further investigations on suitable experimental models are necessary to better describe the profile of *FLAD1* product expression and to establish whether altered flavin homeostasis could be listed among the several factors associated with ALS pathogenesis. Nevertheless, as discussed in a paragraph below, we got the first evidence that altering the expression of FADS can disturb cholinergic transmission [124] in *Caenorhabditis elegans*, a suitable model for studying neuronal dysfunctions.

### 5.2. Rf Muscular Homeostasis and its Alterations

Since energy demand is crucial for cardiac and skeletal muscle, which mostly rely on glucose, fatty acids, and amino acid mitochondrial metabolism, it is not surprising that alterations of flavoenzymes, flavin supply and trafficking principally affect these tissues.

As schematized in Figure 4, mitochondrial respiratory complexes I and II, the Electron Transfer Flavoprotein (ETF) and its Ubiquinone Oxidoreductase (ETF-QO, EC 1.5.5.1) together with several ETF dependent dehydrogenases, is involved in the metabolism of fatty acids and certain amino acids, which are efficient alternatives to glucose as energetic fuel for muscles. Moreover, Succinyl-CoA production, an essential process for ketone bodies utilization in muscle and hearth, also requires the flavin-dependent reaction catalyzed by the mitochondrial enzymatic complex oxoglutarate dehydrogenase (EC 1.2.4.2).

Many years ago, while investigating the mitochondrial bioenergetic alterations in an Rf-responsive patient suffering for MADD (OMIM #231680), we described human *vastus lateralis* muscular mitochondrial flavoproteome and its dependence on Rf supply [84,125]. The proteomic investigation, during the symptomatic phase, revealed a decrease or the absence of several flavoenzymes, related to flavin cofactor-dependent mitochondrial pathways and of mitochondrial or mitochondria-associated calcium-binding proteins. All deficiencies were completely rescued after Rf treatment. Our studies demonstrated for the first time, the previous hypothesis by [126] of a profound involvement of Rf/flavin cofactors in modulating the level of a number of functionally coordinated polypeptides involved in fatty Acyl-CoA and amino acid metabolism, extending the number of enzymatic pathways known to be altered in Riboflavin Responsive (RR)-MADD.

At those times, human genes responsible for flavin cellular and sub-cellular transport, as well as for FAD metabolism were not yet identified. The recent identification of *FLAD1* as a novel mitochondrial disease gene [80] confirmed the crucial role of flavin homeostasis in muscular bioenergetics. Initially identified as MADD, this novel inborn error of metabolism—resulting in Rf responsive and not responsive mitochondrial myopathy—has now been identified as LSMFLAD (lipid storage myopathy due to flavin adenine dinucleotide synthase deficiency, OMIM #255100).

As a consequence of a reduced FAD synthase enzymatic activity in fibroblasts from LSMFLAD patients, a significant reduction of flavin cofactors was detectable at the mitochondrial level lading to a reduced amount of mitochondrial flavoenzymes ETFDH and flavoprotein subunit of succinate dehydrogenase (SDHA), whereas only a slight reduction of flavin cofactors at the cellular level was observed [80]. In fibroblasts from another patient described later, a severe reduction of cellular flavin content was observed, as associated with a drastic reduction of FAD synthase activity [127].

The explanation of the observed residual capability to synthesize FAD, even in the case of frameshift or non-sense “hot spot” mutations in *FLAD1* exon 2 (expected to produce inactive hFADS isoform 1 and 2), derived from the demonstration of the existence of novel shorter transcript variants (Figure 3), whose translation starts beyond the mutated points. One of these transcripts encodes for a protein called isoform 6 or “emergency protein”, which ensures patients’ cells not to be completely deprived of FAD, and therefore still alive [73].

In skeletal muscle biopsies from these patients, besides lipid storage caused by the derangement of fatty Acyl-CoA dehydrogenases, a global decrease of COX and/or SDH histochemical staining was described, as well as multiple respiratory chain enzyme deficiencies measured in the majority of cases tested affecting complexes I, II, III and/or IV with a variable extent. Normal muscle respiratory chain enzyme activities were found in one case. Ragged-red fibers have not been reported, so far [80,128,129].

Thereafter, the number of LSMFLAD patients described at present are continuously increasing [23,80,130] and this prompt us to continue our studies devoted to gain further insight into the still unclear molecular aspects concerning Rf conversion to FAD, using different human cell models, always and keeping in mind the idea that different flavoproteome expression requires different regulation of flavin homeostasis.

Concerning *FLAD1* gene, the main still unsolved problems are the tissue-specific expression profile of *FLAD1* transcripts and the sub-cellular localization of different FADS isoforms, taking into account the fact that FAD synthase must be ubiquitously expressed.

Concerning muscular isoforms of FAD synthase, nothing is known, at the moment, about their expression at the protein level and their sub-cellular localization.

Unpublished data from our laboratory, concerning this point, are shown in Figure 5: in human muscle biopsy for the first-time the simultaneous presence of gene transcript isoforms 1, 3 and 4 are detected; unfortunately, the existence of isoform 2 and 6 transcripts can be neither excluded nor proven for a technical reason. The presence of the transcript isoform 1, as well as the enrichment in muscular lysates of a protein of about 65 kDa (as estimated by SDS-PAGE) [131] allow us to propose that a mitochondrial FAD forming process is present in human muscle, as in other sources [72,81], but a ton of experiments are still necessary to gain further insights into this issue and to close an old debate concerning the mitochondrial localization of FADS [19,80,86,88].

Moreover, we certainly know that human muscle mitochondria can hydrolyze FAD, via an AMP- sensitive FAD diphosphatase (EC 3.6.1._), as mitochondria and nuclei from a different origin, can do, in the so-called Rf/FAD recycling pathway [63,81,84]. Quite interestingly, increased activity of FAD hydrolyzing enzyme and mitochondrial recycling have been associated with mitochondrial flavin level reduction and to an alteration in mitochondrial flavoprotein function [84,132]. However, we still do not know if the FAD destroying activity, we measured in mitochondrial muscle is due to either a Nudix or a not-Nudix hydrolase [133]. The recent discovery that isoform 2 of FAD synthase discloses, under certain experimental conditions, a “hidden” not-Nudix hydrolase face (working as a bi-functional protein) leaves to future investigation the possibility that mitochondrial muscular FADDPase is a product of *FLAD1* gene [34,35].

Another under-investigated point, which necessitates further research to better define Rf homeostasis in muscle, is the expression profile and the sub-cellular localization of Rf transporters. RFVT2 seems to be the principal Rf transporter expressed, as in “The human protein Atlas database” (https://www.proteinatlas.org/). A systematic study concerning this point in the muscle of both control individuals’ patients suffering from Rf-responsive myopathies is still missing (see also [7,130]).

Indeed, a transient form of MADD has been related to deficiencies in RFVT1 (OMIM #615026). Nevertheless, in this case, the muscular symptoms of the newborn child are due to the altered placental distribution of flavin from the mother [134,135,136].

As outlined before, another important component of the mitochondrial FAD homeostasis is the translocator firstly identified in *S. cerevisiae* and named *FLX1* [86,88]. The existence of a human counterpart, nowadays known as the human mitochondrial FAD translocator, SLC25A32 or MFT, was proposed for the first time at the functional level in human muscle as a component of a recycling pathway named Rf/FAD cycle [63,84].

The deficiency of SLC25A32 (OMIM #616839) is causative of an Rf-responsive exercise intolerance (RREI) and presents biochemical features typical of MADD [89,90]. In a skeletal-muscle biopsy obtained from an RREI patient, a faint SDH staining was observed together with some ragged-red fibers [89].

Biochemical assays of complex I, complex II+III, complex IV and citrate synthetase activities was performed in muscle biopsy of another RREI patient as described before. A reduction of complex II and combined II+III activities compared to the control muscle indicated an OXPHOS complex II deficiency. Staining of muscle tissue also revealed for this patient the presence of many ragged-red fibers, as well as multiple cytochrome oxidase-negative muscle fibers [90]. Similar alterations were found in fibroblasts [89,90].

### 5.3. A Molecular Rationale for Mitochondrial Flavoproteome Derangement: the Significance of Rf Therapy

As briefly outlined in Figure 4, mitochondria are plenty of flavoproteins, whose derangements correspond to well-characterized inborn errors of metabolism. From the biochemical (or enzymological) point of view, we can distinguish between defects of β-oxidation, of AA-oxidation, of Krebs cycle components or respiratory chain complexes I and II, all affecting mitochondrial bioenergetics. A survey of all flavoproteins derangements is out of the scope of this review, but we would like to distinguish between single enzyme mutation or multiple enzymatic deficiencies [23,103,137,138].

In the case of a single enzymatic mutation very often the mutation impairs somehow the apo-protein altering the affinity for the flavin cofactor (*k*_m_ or *k*_d_, [139]) and in some cases the folding/stability of the apo-protein [140,141]. Considering these hypotheses, Rf therapy could be quite simply explained as an increase in cofactor availability, which can either compensate the higher *k*_d_ for the cofactor (and therefore the enzymatic activity) or prevent protein misfolding/degradation.

In the case of a multiple flavoproteome derangements, an event which is common to a number of flavoproteins should be altered: (i) transport of vitamin as in the case of RTD (ii) formation of cofactor as in the case of LSMFLAD (iii) regeneration of cofactor to allow the flavoproteome to work as in the case of MADD (iiii) cofactor delivery to nascent holo-flavoproteins [78].

In case of accumulation of unfolded proteins within mitochondria, cells employ a transcriptional response known as the mitochondrial unfolded protein response (UPR(mt)) to promote the repair and recovery of defective mitochondria [142,143].

The possibility that some common transcriptional, post-transcriptional events occur and sense Rf deficiency or flavoproteome derangement presented long ago have been evolved to explain multiple enzymatic derangements both in MADD and in another mitochondrial encephalomyopathy due to alteration of AIF [81,124,143,144]. We also are considering the hypothesis that a redox-epigenetic control, presumably mediated by lysine-demethylase, can start from nuclei [18,81]. More recent literature concerning Rf sensing was elsewhere cited in this review [43,54].

Noteworthy is the case of AIF, a FAD-containing protein with NADH-dependent oxidoreductase activity. Troulinaki group and others demonstrated that AIF physically interacts and stabilizes the oxidoreductase CHCHD4/MIA40, hence assisting the correct biogenesis of the respiratory chain complexes in addition to its death-related role [145,146,147,148,149,150].

Deleterious mutations in the human *AIFM1* gene are associated with rare inherited X-linked mitochondrial disorders [151]. The first deleterious mutation in the *AIFM1* gene was found in two consanguineous infant males, showing progressive mitochondrial encephalomyopathy [151]. As a result, the recombinant mutant AIF protein is structurally unstable, shows aberrant FAD incorporation, and, consequently, impaired redox properties [151,152]. To date, a significant array of mutations in the *AIFM1* gene have been identified in patients showing a wide range of clinical presentations [153,154].

Focusing again on MADD (OMIM #231680), it is a well-known rare autosomal inherited disease associated with an impaired fatty acid, amino acid and choline metabolism resulting in lipid droplets accumulation in skeletal muscles, high plasmatic and urinary levels of acylcarnitine and organic acids and respiratory chain deficiency [155,156].

It is now well-accepted that genetic variants carrying mutations in both the apo-protein of ETF, as well as ETF-QO are responsible for the disease [157,158]. For the former gene, the phenotype is, generally, severe for the latter milder and often sensible to treatment with high doses of Rf. The reader is referred to [130] for a clear distinction.

As outlined in Figure 4, ETF and ETF-QO are mitochondrial flavoproteins functionally primarily connected to β-oxidation. In the case of derangements of these oxidoreductases, the first step of β-oxidation, catalyzed by a different type of Acyl–CoA dehydrogenases cannot work correctly, since they require regeneration of FADH_2_ deriving from oxidation of Acyl groups. Certain flavin-dependent amino acid catabolism and respiratory chain impairment in fibroblasts and muscle biopsies have been described together with alteration of ROS homeostasis and ATP shortage.

All these biochemical abnormalities were exhaustively described in cell contests [126,159,160,161]; model organism used to describe MADD are summarized in Table 1, at the end of this review, together with metabolic dysfunctions induced by gene mutations.

The novel important concept emerging from the latest literature is that a secondary derangement of flavin homeostasis can be induced by damage of electron flux via ETF/ETF-QO, thus generating a sort of “flavin vicious cycle”.

## 6. Model Organisms to Study Flavin Homeostasis Alterations

In order to better establish the biochemical alterations produced by defects in flavin metabolic pathways, two models have been initially introduced in our and other laboratories: the yeast *Saccharomyces cerevisiae* and the nematode *Caenorhabditis elegans*. More recently, murine and fruit fly *Drosophila melanogaster* models have been also introduced in the study of flavin linked neuromuscular disorders.

### 6.1. Saccharomyces Cerevisiae

The yeast *Saccharomyces cerevisiae* has long been used as a eukaryotic model organism mostly due to its simple and quick genetic manipulation. In this organism, the first genes involved in flavin metabolism were identified, by functional genomics approaches and cloned. Once a clear parallel to mammalian genes and biochemical processes involved in vitamin homeostasis was established, the yeast model was also useful to identify by complementation the human genes correlated to flavin homeostasis.

Besides the conservation of common fundamental processes and the significant homology of the cellular flavoproteome [162], it should be mentioned that profound differences exist between these lower eukaryotes and men, as far flavin homeostasis is concerned.

First of all, differently from mammals, yeasts, as well as fungi, plants and bacteria, have the ability either to synthesize Rf *de novo* or to take it from outside [163,164]. All yeast enzymes required for de novo Rf biosynthesis are encoded by the *RIB* genes (*RIB1, RIB2, RIB3, RIB4, RIB5*, and *RIB7*) [165], which of course have no counterpart in mammals.

Then, Rf uptake in *S. cerevisiae* is mediated by the product of *MCH5* gene [166], which has no counterpart in mammals, since RFVTs are evolutionary recent. Mch5p is a high-affinity transporter (*k*_m_ = 17 µM) with a pH optimum at pH = 7.5, operating by a facilitated diffusion mechanism. The expression of *MCH5* is regulated by cellular Rf content. Thus, *S. cerevisiae* has a mechanism to sense Rf and avert Rf deficiency by increasing the expression of the plasma membrane transporter [166].

In the *S.*
*cerevisiae* genome, the first genes encoding for monofunctional RFK and FADS were identified and named *FMN1* [167] and *FAD1* [168], respectively. As mammals, yeasts use two different enzymes for FAD production, conversely, most prokaryotes depend on a single bifunctional enzyme [169,170,171].

*FMN1* encodes for a protein of 24.5 kDa which shows sequence and structure similarity to the RKF-module of prokaryotic FADS and appears largely conserved through evolution up to the human enzyme [7]. Immunoblotting analysis of yeast subcellular fractions revealed that Fmn1p is localized in microsomes and in mitochondria [167].

Fad1p, the sole known protein isoform generated by *S. cerevisiae FAD1* gene, is a 35.5 kDa, soluble enzyme, essential for yeast life, whose crystal structure was solved in a complex with FAD in the active site [172]. Fad1p is a single-domain monofunctional enzyme containing the sole PAPS or FADSy domain, which has little or no sequence similarity to the prokaryotic FAD-forming enzymes. This feature makes prokaryotic FADS a good candidate for novel antibiotics [31,173].

Homology searching using yeast Fad1p as a query leads to the identification of the human *FLAD1* gene [31], therefore opening the possibility to study the molecular mechanisms underlying LSMFLAD myopathies. As discussed above, the human protein isoforms are more numerous and structurally complicated than their yeast monofunctional counterpart, as the longest human isoforms are composed of two fused domains (see above, Figure 2 and [31,71,79]). Interestingly, the yeast protein Fad1p strongly resembles the recently discovered human isoform 6 (hFADS6): the two proteins exhibit 32% amino acid identity.

The mammalian FADHy domain has, indeed, a single domain enzymatic counterpart in yeast, namely the product of *FPY1,* which performs a non-Nudix diphosphatase activity [77] capable of hydrolysing in vitro FAD, NAD(H), and ADP-ribose in presence of K^+^ and divalent cations. Indeed, our group observed that *S. cerevisiae* mitochondria (SCM) are able to perform FAD hydrolysis via an enzymatic activity which is different from the already characterized Nudix hydrolases; since this activity is regulated by the NAD redox status, an experimental link between NAD and FAD mitochondrial world has been proposed [174].

While there is no doubt about a mitochondrial localization for Fmn1p [86,167], the existence of a mitochondrial FADS isoform in yeast is still controversial. First, it was reported that FAD is synthesized by Fad1p exclusively in the cytosol, since no mitochondrial targeting signal is present in this protein [86,168], but later on, mitochondrial FAD-forming activity was revealed in yeast mitochondria [88,144,175].

The yeast mitochondrial transporter of FAD was identified by Tzagoloff group [86], who named *FLX1* the encoding gene; the transporter was proposed to catalyze and exchange between flavins across the mitochondrial membrane, with cytosolically synthesized FAD entering in exchange with mitochondrial FMN. The role of *FLX1* in catalyzing intra-mitochondrial FAD efflux to the cytosol was proposed by our group [88]. *S. cerevisiae* flx1 mutant strain was used to identify, by complementation, the human orthologue *SLC25A32A* [87,91].

Despite the unsolved controversies about the direction of FAD transport catalyzed by Flx1p in yeast, this mitochondrial translocator certainly has a role in maintaining the mitochondrial flavoproteome [162]. Flx1, null and mutated strains, were the first models used to mimic alterations of FAD homeostasis and their relationships with derangements of mitochondrial flavoprotein biogenesis [88] as that found in human MADD [87] or, now, better RREI [89,90].

The enzymatic activities of certain mitochondrial FAD-binding enzymes, i.e., lipoamide dehydrogenase and Sdh1p are altered in *S. cerevisiae* strain lacking *FLX1* [86,88], which resulted in a small colony and also a respiration-deficient phenotype [144,175]. In another set of studies, the loss of function of the SLC25 family member Flx1 was coupled to the loss of function of Hem25: this resulted in a respiratory-deficient phenotype, indicative of mitochondrial impairment [176].

Unfortunately, other flavoenzymes, which are expected to be altered in MADD/LSMFLAD/RREI, were not assayed in yeast models. In particular, data are missing concerning the system ETFα/ETFβ/ETF-QO, which are mitochondrially located and encoded by *AIM45*, *CIR1* and *CIR2* in *S. cerevisiae*, whose alteration can be a cause of ROS unbalance [162,177,178]. Conversely, the activity/level of lipoamide dehydrogenase—a component of enzymatic complexes that decarboxylate pyruvate, oxoglutarate and oxoacids deriving from branched-chain amino acids—was not tested, as well as that of ETF/ETF-QO has not yet been investigated in RREI patients’ cells.

The deletion of *FLX1* was accompanied by a significant ATP shortage and ROS unbalance in glycerol-grown cells [175]. Moreover, the *flx1∆* strain showed H_2_O_2_ hypersensitivity and decreased lifespan [175].

In biochemical and cellular alterations induced by altered mitochondrial FAD transporter, a central role is played by the flavoprotein subunit of SDH. This is true not only in yeast, but also in human cell model recently introduced [91].

According to our hypothesis, flx1p can act as a nutrient sensor, modulating Sdh1p biogenesis via a post-transcriptional control, that involves in a sort of “retrograde control” putative regulatory sequences located in the UTR region upstream the *SDH1* coding sequence [144]. Sdh1p biogenesis can also be profoundly altered by mutations of ancillary proteins required for covalent FAD insertion into the apo-protein [179]. The reader is referred to an excellent review on this topic [180].

Using *S. cerevisiae* as a model to study flavoproteome derangements linked to alteration of Rf homeostasis, our group get the first bioinformatic evidence that the so-called transcriptional “flavin network” hypothesized in human muscle [125] can actually exist in yeast [175]. A search for possible cis-acting consensus motifs in the regulatory region upstream SDH1-ORF revealed the presence of two protein-binding motifs, which are conserved also in the regulatory region of genes encoding for proteins involved in flavin homeostasis. The two putative transcriptional regulators are Msn2/4 (stress response factor) and Rox1p (involved in the regulation of the expression of proteins involved in oxygen-dependent pathways, such as heme biosynthesis). As discussed in [175], this finding strengthens the well-described relationship between oxygen/heme metabolism and flavoproteins. These elements might coordinate a response to the metabolic changes induced by altering flavin homeostasis, having as a consequence a change in the level of mitochondrial succinate, a potential epigenetic regulator [174,175].

If flavoproteome derangements due to the mutation in *FLAD1* or *SLC25A32* in humans is due to the lack of intramitochondrial cofactor, affecting apo-flavoprotein assembly/stability and inducing mitochondrial unfolding protein response [143] or, rather, affecting the signaling pathway introduced as the “flavin network” in yeast [175] is still matter of future investigation.

Finally, it should be noted that some limitations on the use of *S. cerevisiae* to mimic MADD/LSMFLAD consists in the observation that (i) *FAD1* gene deletion is lethal and (ii) β-oxidation is carried out only in the peroxisomes in yeast [181]. Other models are more suitable for these human inborn errors of metabolism, as for example worms.

### 6.2. Caenorhabditis Elegans

*C. elegans* is one of the best model organisms in biology research, since in addition to the easy manipulation, it possesses physiological and pharmacological properties common to those of higher animals. Moreover, this nematode has been completely defined with respect to anatomy, genetics, development, differentiation and behavior.

*C. elegans* is easily maintained in the laboratory on agar plates or in a liquid medium using *Escherichia coli* as a food source. Its life cycle from egg to adult takes about 3 days at 22 °C. The first-stage larva hatches from the egg and proceeds through three additional stages of larval development, before reaching reproductive maturity as an adult [182]. Finally, *C. elegans* was the first multicellular organism for which the genome was completely sequenced [183]. Thus, the use of this simple and inexpensive model seemed appropriate to study the relationships between FAD synthesis and flavoprotein biogenesis.

*C. elegans*, like humans, is not able to synthesize Rf by itself, therefore its only source is the food, mainly based on bacteria. Vitamin absorption occurs in the intestine through transport systems that introduce Rf into intestinal cells.

Two potential riboflavin transporters orthologs of the human riboflavin transporters were identified in *C. elegans*, Y47D7A.16 (*rft-1*) and Y47D7A.14 (*rft-2*) which share 33.7 and 30.5% identity, respectively, with hRFVT3 [184].

*rft-1* is localized on Chromosome V and encodes a single protein isoform (RFT-1) of 427 aa, predicted to have 11 TM domains with an extensive intracellular loop between the sixth and seventh TM domains [184]. It is expressed in the intestine and a small subset of neuronal support cells along the entire length of the animal and it has been proven to be important in embryonal development. The mediated Rf transport has an acidic pH dependence, saturability (apparent *k*_m_ = 1.4 ± 0.5 µM), inhibition by Rf analogs, and Na^+^ independence. The expression of *rft-1* is suggested to be adaptively regulated by extracellular Rf levels. [184].

*rft-2* is localized on Chromosome V that generates two alternatively spliced mRNAs namely *rft-2a* and *rft-2b,* which encode for two putative protein isoforms (RFT-2a and- b) of 463 and 476 aa, respectively. Further details on structural predictions of these two isoforms are reported in [184,185]. At the moment other three putative shorter isoforms are reported in WormBase (https://wormbase.org/species/c_elegans/gene/WBGene00021626#0-9f-10). *rft-2* is expressed mainly in the intestine and pharynx [184]. No data exist about a possible subcellular localization. It is involved in maintaining the body homeostatic Rf levels at the whole animal level all through its life as demonstrated by studying the expression pattern of *rft-2* at different life stages. The abundance of *rft-2* transcript in the whole animal was upregulated in Rf-deficient conditions (10 nM) and downregulated at high doses of Rf supply (2 mM) as compared with control (10 μM). The expression of *rft-2* was found to be adaptively regulated in vivo when transgenic worms were maintained under different extracellular Rf levels, which was also mediated partly via changes in the *rft-2* levels that directs towards the possible involvement of transcriptional regulatory events [185].

Concerning the intracellular conversion of Rf in its cofactors, little is known about flavokinase encoded by R10H10 gene in *C. elegans*. The R10H10 protein deduced from the cDNA sequence contains 135 aa with an estimated molecular mass of 14.7 kDa and shows 46.7% identity with the human protein.

Homology searching in *C. elegans* databases using hFLAD1 as template lead to the identification of the orthologue worm gene i.e., *flad-1* (R53.1), which is organized in 8 exons and located on chromosome II. The two products from *flad-1* transcription are trans-spliced and generate two proteins, which show 37% identity and 55% similarity to the human homologues [124].

Currently, in WormBase (https://wormbase.org/species/c_elegans/gene/WBGene00011271#0-9f-10) other two transcripts coding for two shorter proteins (containing the sole PAPS or FADSy domain) which show 38% identity to the human isoform 6.

Two worm models, linked to the alteration of flavin homeostasis with possible neuromuscular and neurodegenerative consequences, have been reported up to now [124,184].

The first model, introduced by our group, resulted from the transient silencing of the *flad-1* gene by feeding [124] causing a 50% reduction of total flavin content. Phenotypical changes, among which reduced proliferation rate and impairment in locomotion behavior were observed in interfered nematodes. Besides decreased respiratory activities, ATP shortage and the reduction in the activity of several flavoenzymes, as SDH and glutathione reductase were observed in *flad1*-interfered nematodes. In this aspect, *flad-1* interfered nematodes in our opinion proved to be a suitable animal model system for studying human pathologies with alteration in flavin homeostasis/flavoenzyme biogenesis.

In many mitochondrial mutant nematodes, altered mitochondrial respiration can enhance oxidative stress and induce the formation of free radicals [186]. Thus, we assessed reactive oxygen species (ROS) levels in flad-1-silenced animals, observing a significant increase, which presumably might cause a cellular stress response.

Proteomic studies in interfered nematodes revealed that at least 15 abundant proteins are affected by *flad-1* gene silencing, some of which are not flavoproteins, possibly confirming the transcriptional/post-transcriptional control exerted by FAD shortage, on mitochondrial protein expression, as already discussed for yeast [144,175] and for the co-ordinated network of proteins in RR-MADD patients [125,143,187].

Another consequence of *flad-1* silencing in worms is linked to altered locomotion behavior, together with a possible reduction in neurotransmitter production, the measured alteration in aldicarb sensitivity, suggested a possible specific derangement of cholinergic transmission secondarily connected to alteration of flavin homeostasis [124]. Experiments are currently going on in our laboratory both in worm and in human cell models to further describe the biochemical and functional consequences of inducing FAD shortage in neurons.

The second worm model was created in Said laboratory, to provide an animal model to study pathologies associated with mutations of human RFVT3, concerns the silencing of the worm plasmatic Rf transporters [184]. The knockdown of each single or both transporter genes by RNAi resulted in reduced fertility in worms. A high concentration of exogenous Rf moderately reverted the induced Rf deficiency. Unfortunately, these strains were not characterized at the biochemical level.

More recently, Rf level in nematodes has been also critically associated with food uptake and foraging behavior. These effects are due to the regulation of specific protease gene expression and intestinal protease activity in a TORC1 mediated way, which senses ATP shortage due to altered FAD production [188].

As stressed above, the proper mitochondrial activity and ATP production is also ensured by the regulatory role of flavoprotein AIF, which shuttles between mitochondria and nucleus in the caspase-independent apoptosis pathway. The worm AIF homolog WAH-1 is a mitochondrial protein that undergoes conformational changes according to the redox status of the surrounding milieu. Consistently *wah-1* downregulation (mimicking human X-linked encephalomyopathy) in *C. elegans* compromises oxidative phosphorylation and reduces lifespan [145].

Notably, in agreement with the hypothesis of coordinated transcriptional regulation of flavoproteome [125,143,175] a nuclear-encoded mitochondrial stress signaling pathway was identified in these worm models [142] involving activation of *hsp-6* (heat-shock-protein-6), *gst-4* (glutathione-S-transferase), and *sod-3* (superoxide dismutase) promoters, associated with the enhanced expression of the HIF-1 target gene *nhr-57* [145].

Despite the presence of putative FAD and NADH binding domains, it seems that WAH-1 does not incorporate cofactors. This may be a reason why the effect of Rf treatment could, unfortunately, not be established in this model.

### 6.3. Mouse and Drosophila Melanogaster

Recently further animal models have been introduced in order to elucidate the RFVT contribution to motor neuron differentiation and the pathogenic mechanisms of BVVLS [117,189,190].

Riboflavin transporters, RFVT2/*Slc52a2* and RFVT3/*Slc52a3*, have been identified in rodents. mRFVT2 is the orthologue of both hRFVT1 and hRFVT2 and it is involved in hepatic homeostasis of Rf in mice [38]. mRFVT3 is the orthologue of hRFVT3 [190] and it has been shown to be the main transporter involved in carrier-mediated RF uptake in the native mouse small and large intestine [191]. Experiments performed on mouse colonoids demonstrated that mRFVT3 is up-regulated both at transcript and protein level after NaB treatment [55]. Further, it has been shown to be essential for mouse development, with *Slc52a3* deficiency resulting in early embryonic lethality associated with defects in placental formation, but dispensable for neural differentiation and short-term maintenance [190], suggesting that in BVVLS patients with mutations in RFVT3 alternative transporters may act during embryogenesis to allow full-term development. The physiological role of RFVT3 has been also investigated in [189], in which *Slc52a3* knockout mice have been used. The disruption of *Slc52a3* gene caused neonatal mortality with hyperlipidemia and hypoglycemia owing to Rf deficiency.

To overcome the issue of early mortality in mice, which precludes phenotypic analysis at later developmental stages, [117] turned to the fruit fly *Drosophila melanogaster* as novel in vivo model of BVVLS. Comparative BLAST analysis lead to the identification of *Drosophila SLC52A3* homologous gene, i.e., cg11576. This gene has been named *drift* (Drosophila riboflavin transporter) and encodes for a protein that exhibits 36.9% amino acid identity with hRFVT3 and it is expressed in several adult tissues as head, gut, abdomen and thorax. The knockdown of *drift* revealed reduced levels of Rf, downstream metabolites, and electron transport chain complex I activity, which in turn resulted in abnormal mitochondrial membrane potential, respiratory chain activity and morphology. *drift* knockdown also resulted in severely impaired locomotor activity and reduced lifespan. These phenotypes have been partially rescued using riboflavin-5′-lauric acid monoester (RLAM), a novel esterified derivate of Rf.

In conclusion, it is well assessed that fine coordination among Rf supply, flavin cofactor homeostasis and apo-flavoproteome maintenance is necessary for efficient cellular bioenergetics. Novel experimental models are, therefore, needed to address the huge number of molecular processes and transcriptional and post-transcriptional networks not yet elucidated underlying the muscular and neuronal flavin-dependent mitochondrial oxidative pathways. This is pivotal not only for a better understanding of the molecular rationale of Rf therapy in responsive patients, but, hopefully, also in the aim to potentiate possible therapeutic intervention in Rf-related neuromuscular diseases, in cases, unfortunately, not responding to vitamin therapy.

**Table 1 ijms-21-05310-t001:** Model organisms to study MADD.

Model Organism	Gene Mutation	Metabolic Dysfunction	References
*Caenorhabditis elegans*	*let-721*	*let-721* mutants are maternal effects either lethal or semi-sterile.	[192]
*Drosophila Melanogaster*	*Etfdh*	Biochemical defects observed in the severe forms of MADD: embryonic accumulation of short-, medium and long-chain acylcarnitines, ETF-QO activity markedly decreased, impaired cofactor association via structural destabilization and consequently enzymatic inactivation.	[193]
Zebrafish	*Etfdh*	Metabolic and mitochondrial dysfunctions, alteration of plasma acylcarnitine and organic acid profiles, reduced oxidative phosphorylation, increased glycolytic flux and the upregulation of the PPARγ-ERK pathway associated to aberrant neural proliferation and motility defects.	[194]
	*Etfa*	Pathological and biochemical features similar to those observed for MADD affected individuals, including brain, liver and kidney diseases. An increased signaling of the mechanistic target of rapamycin complex 1 (mTORC1) responsive to treatment with rapamycin was also found.	[195]
Mouse	*Etfdh^(h)A84T^*	First RR-MADD mouse model with an *Etfdh* (h)p.84A > T mutation. The mice, initially normal, developed the clinical and biochemical features typical of MADD under high fat and Rf deficiency diet. Tissues from these mice exhibited a significant decrease of both FAD concentration and ETFDH protein level, which were ameliorated by Rf treatment.	[196]

## Figures and Tables

**Figure 1 ijms-21-05310-f001:**
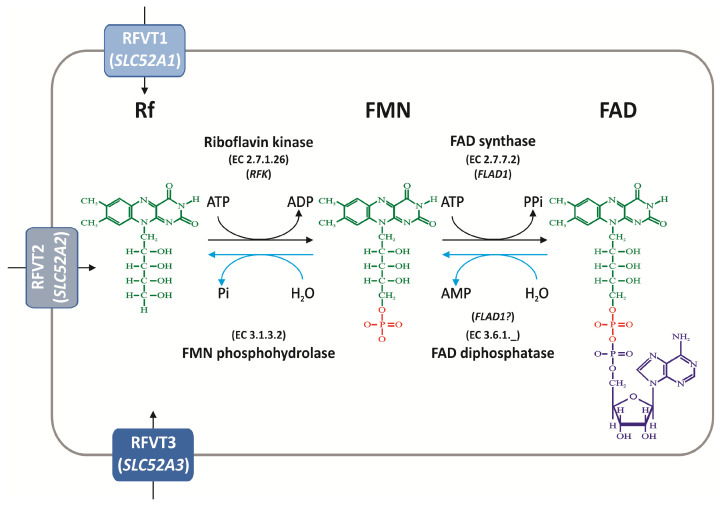
Chemical structures and metabolic conversion of Riboflavin (Rf) and flavin cofactors in man. The chemical structures of Rf, flavin mononucleotide (FMN) and flavin adenine dinucleotide (FAD) and the enzymes involved in the conversion of Rf to FAD and vice versa are reported, together with the corresponding genes’ names. In humans Rf is taken up in a carrier-mediated process by three transporters named Rf transporter 1 (RFVT1), RFVT2 and RFVT3 (encoded by *SLC52A1-3*). Inside the cells, Rf conversion to flavin cofactors occurs in two steps catalyzed by Riboflavin Kinase (encoded by *RFK*) forming FMN, and FAD Synthase (encoded by *FLAD1*) forming FAD. Recycling of the vitamin can move from FAD in two steps, catalyzed by FAD diphosphatase and FMN phosphohydrolase.

**Figure 2 ijms-21-05310-f002:**
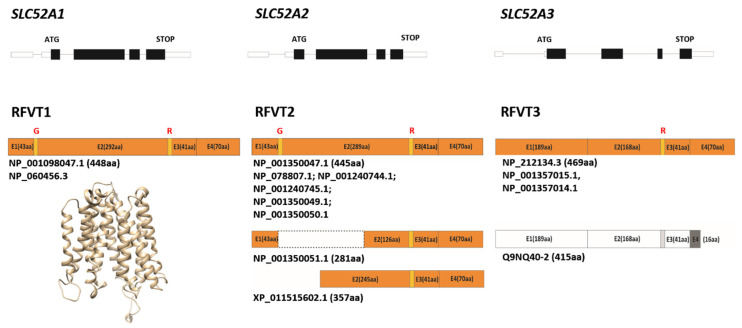
Sketch representation of RFVT protein isoforms. At the top of the figure, schematic representation of genes encoding for RFVT1, RFVT2 and RFVT3. Exons are represented as black boxes, introns as lines between exons and UTRs as short white boxes. In the lower part of the figure, schematic representation of RFVT protein isoforms. In orange RFVT isoforms as reported in NCBI; in white a RFVT3 isoform, as reported in [43]. Length as coded amino acids for each exon is reported (in brackets). Under each isoform, accession number and length for each protein. The homology structural model of hRFVT1 was built, using as a template the equilibrative nucleoside transporter 1 (PDB code 6OB7) using the SWISS-MODEL software. The protein shows the 11 transmembrane α-helical segments nearly parallel to the membrane axis.

**Figure 3 ijms-21-05310-f003:**
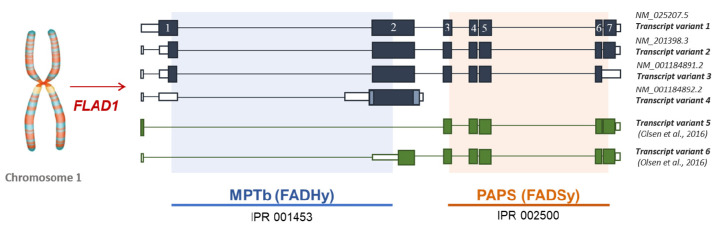
Sketch representation of *FLAD1* transcript variants. In dark blue are indicated the four transcript variants reported in RefSeq whereas in green are indicated the two *FLAD1* novel transcript variants as reported in [80]. Big colored boxes represent coding regions while lines represent introns. In transcript variant 4 light colored boxes represent different coding regions with respect to the transcript variant 1. The two functional domains of the longest isoforms are colored in the background. Homology modeling of the single domains are reported in [7,34,73,74].

**Figure 4 ijms-21-05310-f004:**
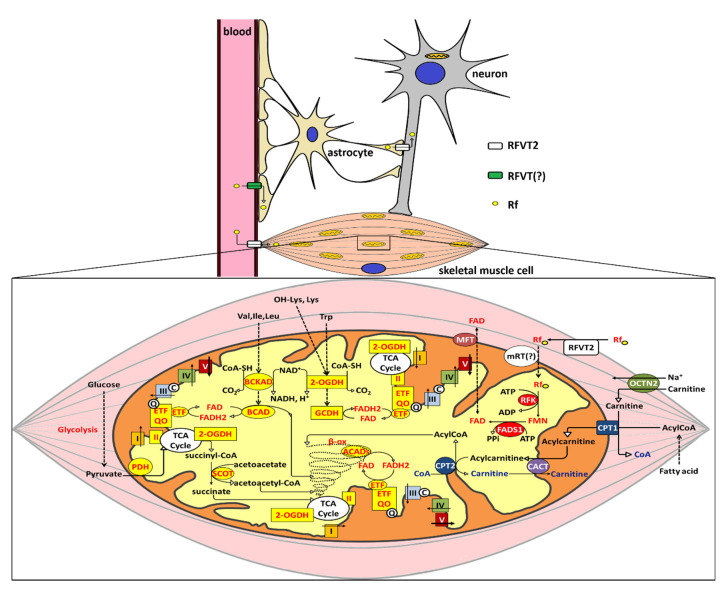
Rf transport and FAD mitochondrial delivery in skeletal muscle and neuronal cells, in relationships with flavin-dependent mitochondrial oxidative pathways. At the top of the figure, the Rf uptake from blood into target cells (neuronal and skeletal muscle cells) presumably involving RFVT2. At the bottom of the figure, a magnification of mitochondrial flavoprotein metabolism in muscle is depicted. Flavoproteins are depicted in yellow. Abbreviated names of transporters and enzymes are indicated in the figure as listed below: RK: Riboflavin kinase (EC 2.7.1.26); RFVT2: plasma membrane riboflavin transporter 2; FADS1: FAD synthase isoform 1 (EC 2.7.7.2); mRT: mitochondrial riboflavin transporter; MFT: mitochondrial folate transporter; CACT: mitochondrial carnitine acylcarnitine translocase; OCTN2: organic cation transporter novel 2; CPT1: carnitine palmitoyltransferase 1; CPT2: carnitine palmitoyltransferase 2; I: respiratory chain complex I (NADH-ubiquinone oxidoreductase); II: respiratory chain complex II (succinate dehydrogenase); III: respiratory chain complex III (ubiquinol-cytochrome c reductase); IV: respiratory chain complex IV (cytochrome c oxidase); V: respiratory chain complex V (ATP synthase); 2-OGDH: 2-oxoglutarate dehydrogenase complex; ACADs: Acyl-CoA dehydrogenase various isoforms; BCKAD: branched-chain α-keto acid dehydrogenase complex; ETF: electron transfer flavoprotein; ETF-QO: electron transfer flavoprotein-ubiquinone oxidoreductase; GCDH: glutaryl-CoA dehydrogenase; BCAD: Acyl-CoA dehydrogenase branched chain specific (2-methyl-butyryl-CoA dehydrogenase); PDH: pyruvate dehydrogenase; SCOT: succynil-CoA:3-ketoacid-coenzyme A transferase; TCA Cycle: tricarboxylic acid cycle; β-ox: β-oxidation of fatty acids.

**Figure 5 ijms-21-05310-f005:**
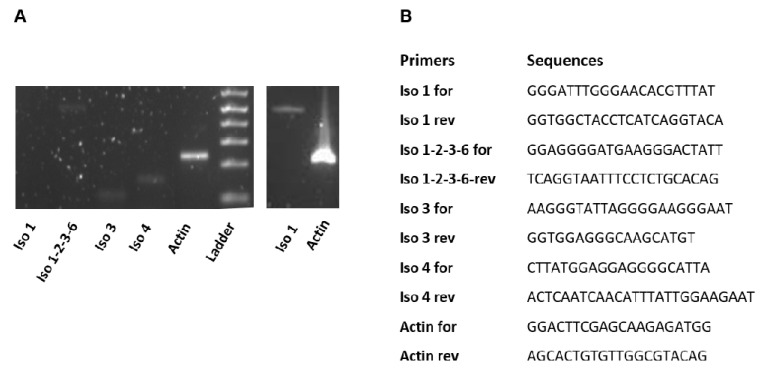
Expression of *FLAD1* transcript variants in human muscle biopsy. (**A**) Total RNA from muscle was prepared to generate cDNA. 75 ng (left panel) or 200 ng (right panel) cDNA prepared from muscle were used as a template for PCR (35 cycles). (**B**) Specific primer pairs were used for the amplification of the transcripts of isoform 1, -3 or -4. No specific primer pairs could be designed for isoforms 2 and -6. Actin mRNA level was used as an internal standard.

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
