# Peer review of "Development of Novel Experimental Models to Study Flavoproteome Alterations in Human Neuromuscular Diseases: The Effect of Rf Therapy"

_ijms, 2020, doi:10.3390/ijms21155310_

Round 1

Reviewer 1 Report

The review entitled “Development of novel experimental models to study flavoproteome alterations in human neuromuscular 3 diseases: the effect of Rf therapy” is a very valuable review where the authors have addressed Rf transporters and the already known as flavin network involved in intracellular flavoproteome maintenance.  The authors deal with different models to study potential therapeutic solution that involved flavoproteins that play a role in neuromuscular diseases.

Anyway, it is possible to contribute to this work with comments and possible improvements in some parts:

- Lines 103-106:…Therefore, extracellular diphospho- (EC 3.6.1._) and monophospho-hydrolases (EC 3.1.3.2) must reform the vitamin, in the brush border, before its delivery into enterocytes and, later on, into each specialised cell of peripheral tissues

Does it mean that FMN and FAD have to be converted into Rf again before being delivered into enterocytes? Do these cofactors come from the thin intestine, as a result of a digestion process? This sentence could be rephrased to be understood better. 

- Lines 128-131: Authors report on the expression profile of SLC52A members in different tissues, naming placenta, intestine and testis. What about this expression in hepatocytes? Are there studies in where the expression of SLC25As in this type of cells has been also reported?

-Line 194 No data are at the moment available on the promoter region of SLC52A2. Can the authors give reasons why this fact?

- In my opinion, the text of the review ends abruptly (lines 846-848).  It seems several sentences are missing since the way to end is weird.

In addition, minor considerations of typing mistakes, lack of resolution and confusing usage of similar terms should be taken into consideration by the authors:

- Throughout the text, the article “the” is used or missing when referring to mRFVTs Could the authors regularize this usage or not of the article?

- Figure 1. The quality of the figure could be improved. As far as my review manuscript is concerned, the low resolution is especially notorious in chemical structures and labels.

- Lines 34-35 (page 1). Did the authors want to say “in the context of” instead of “in the contest of”?

- Line 128 and lines 196-197:  SLC52A and RFVT refer to the same protein so they are used in the text like synonymous of each other. Can this interchangeable use of SLC52As an RFVTs terms lead to confusion? For instance, in figure 2 it is said SLC52As protein isoforms but labels show RFVTs

- Lines 202-203. Adding the pdb CODE 6OB7 would contribute to give complete information about the template used to generate the homology structural model.

- Lines 244, 245  FMN:ATP adenylyl transferase (FADS or FMNAT) in humans are coded by two distinct genes in humans, named as reported in Figure 1. 

It is repetitive. Consider deleting the second “in humans”

- Line 367 total flavin concentration is reported to be ~ 0.1 μM with ~ 0.02 μM. The word must be “to” instead “with”

- Lines 381,382 Two typical very close. Consider changing one “typical” for a synonymous like “average, characteristic, representative, standard…”

- Figure 4. Flavoproteins involved in skeletal muscle cell metabolism and relationships with oxidative pathways. Could the neuron and astrocyte be named at the foot of the figure relating them with the skeletal muscle cell?

- Line 441 Mitochondrial flavoprotein metabolism are depicted. Consider changing the form of the verb: “are” for “is”    

- Lines 655, 656: As discussed above the human protein isoforms are more numerous and structurally complicated that their yeast monofunctional counterpart.  

Since it is a comparison, “that” should be replaced by “than”

- Line 748 (apparent Km = 1.460.5 mM), I can not understand this number. Use the adequate separators (a decimal dot, and a comma to separate thousands) (i.e. 1,460.5 mM if that is the case)

Author Response

Comments and Suggestions for Authors

The review entitled “Development of novel experimental models to study flavoproteome alterations in human neuromuscular 3 diseases: the effect of Rf therapy” is a very valuable review where the authors have addressed Rf transporters and the already known as flavin network involved in intracellular flavoproteome maintenance.  The authors deal with different models to study potential therapeutic solution that involved flavoproteins that play a role in neuromuscular diseases.

Anyway, it is possible to contribute to this work with comments and possible improvements in some parts:

- Lines 103-106:…Therefore, extracellular diphospho- (EC 3.6.1._) and monophospho-hydrolases (EC 3.1.3.2) must reform the vitamin, in the brush border, before its delivery into enterocytes and, later on, into each specialised cell of peripheral tissues

Does it mean that FMN and FAD have to be converted into Rf again before being delivered into enterocytes? Do these cofactors come from the thin intestine, as a result of a digestion process? This sentence could be rephrased to be understood better. 

Done. This sentence has been rephrased.

- Lines 128-131: Authors report on the expression profile of SLC52A members in different tissues, naming placenta, intestine and testis. What about this expression in hepatocytes? Are there studies in where the expression of SLC25As in this type of cells has been also reported?

As far as we know no literature exists on RFVT expression profile in human hepatocytes; conversely Ref. 38 in the revised version reports studies performed in mouse. Indeed, we thank very much the reviewer for his/her comment that prompted us to check in “The human protein Atlas database” as reported at line 120 page 3 in the revised version, according to which RFVT2 seems to be the principal Rf transporter in liver.

Checking again the existing current literature on this point, we found a paper concerning Rf homeostasis in pancreas cells that we added as Ref. 37.

-Line 194 No data are at the moment available on the promoter region of SLC52A2. Can the authors give reasons why this fact?

In the paper (Ref. 54) a transcriptional control of RFVT2 was clearly demonstrated but since RFVT3 is the predominant and most active RF transport system in the intestine, the authors focused only SLC52A3 promoter for characterizing at molecular and experimental level the cis acting regulatory regions.

- In my opinion, the text of the review ends abruptly (lines 846-848).  It seems several sentences are missing since the way to end is weird.

We added a paragraph at the end of section “Model Organisms to study flavin homeostasis alterations”. We thank the reviewer for his/her ameliorating suggestion.

In addition, minor considerations of typing mistakes, lack of resolution and confusing usage of similar terms should be taken into consideration by the authors:

- Throughout the text, the article “the” is used or missing when referring to mRFVTs Could the authors regularize this usage or not of the article?

Done. We deleted the article throughout the entire paper.

- Figure 1. The quality of the figure could be improved. As far as my review manuscript is concerned, the low resolution is especially notorious in chemical structures and labels.

Done. We improved the quality of the figure.

- Lines 34-35 (page 1). Did the authors want to say “in the context of” instead of “in the contest of”?

Done. We corrected with “context”, sorry for this.

- Line 128 and lines 196-197:  SLC52A and RFVT refer to the same protein so they are used in the text like synonymous of each other. Can this interchangeable use of SLC52As an RFVTs terms lead to confusion? For instance, in figure 2 it is said SLC52As protein isoforms but labels show RFVTs.

We thank very much the reviewer for his/her ameliorative suggestion. In the revised version we indicated proteins as RFVTs and coding gene as SLC52As.

- Lines 202-203. Adding the pdb CODE 6OB7 would contribute to give complete information about the template used to generate the homology structural model.

Done. We added the PDB code.

- Lines 244, 245  FMN:ATP adenylyl transferase (FADS or FMNAT) in humans are coded by two distinct genes in humans, named as reported in Figure 1. 

It is repetitive. Consider deleting the second “in humans”

Done. We deleted the second one, sorry for this.

- Line 367 total flavin concentration is reported to be ~ 0.1 μM with ~ 0.02 μM. The word must be “to” instead “with”

Done. We better explained the sentence.

- Lines 381,382 Two typical very close. Consider changing one “typical” for a synonymous like “average, characteristic, representative, standard…”

Done. We used “characteristic” instead of the second “typical”.

- Figure 4. Flavoproteins involved in skeletal muscle cell metabolism and relationships with oxidative pathways. Could the neuron and astrocyte be named at the foot of the figure relating them with the skeletal muscle cell?

Done. We did minimal changes in the figure and rewrote the top part of the legend.

- Line 441 Mitochondrial flavoprotein metabolism are depicted. Consider changing the form of the verb: “are” for “is”.

Done. We corrected the verb, sorry for this. 

- Lines 655, 656: As discussed above the human protein isoforms are more numerous and structurally complicated that their yeast monofunctional counterpart.  

Since it is a comparison, “that” should be replaced by “than”

Done. We corrected the word.

Line 748 (apparent Km = 1.460.5 mM), I can not understand this number. Use the adequate separators (a decimal dot, and a comma to separate thousands) (i.e. 1,460.5 mM if that is the case)

Done. We corrected the number.

Some other linguistic corrections have been independently added.

Submission Date

04 July 2020

Date of this review

15 Jul 2020 12:49:35

Reviewer 2 Report

The review article presented by Tolomeo et al., has discussed about riboflavin transporters along with the molecular rationale of riboflavin therapy in neuromuscular diseases. I feel like this review may have a good impact on the study in this field of medicine in near future. Moreover, this review is very clearly written and well presented. I have one minor suggestion to the author to discuss a little more about, the role of recently identified interactive protein (TMEM237) for RFVT-3 (SLC52A3) transporter (Sabui et al, 2019, AJP-cell Physiology) in the section 3.2. “Rf transporters: what else” (Page 5) and corelate the interactive protein’s role with the effect of Sodium butyrate and TNF alpha on RFVT-3 expression and functionality. I would recommend acceptance of this review with this minor correction.

Author Response

Comments and Suggestions for Authors

The review article presented by Tolomeo et al., has discussed about riboflavin transporters along with the molecular rationale of riboflavin therapy in neuromuscular diseases. I feel like this review may have a good impact on the study in this field of medicine in near future. Moreover, this review is very clearly written and well presented. I have one minor suggestion to the author to discuss a little more about, the role of recently identified interactive protein (TMEM237) for RFVT-3 (SLC52A3) transporter (Sabui et al, 2019, AJP-cell Physiology) in the section 3.2. “Rf transporters: what else” (Page 5) and corelate the interactive protein’s role with the effect of Sodium butyrate and TNF alpha on RFVT-3 expression and functionality. I would recommend acceptance of this review with this minor correction.

We thank very much the reviewer for his/her useful suggestion. In the revised version we added, as requested, a short discussion on this point and, also, we added Ref. 57 concerning TMEM237 molecular function and expression.

Submission Date

04 July 2020

Date of this review

15 Jul 2020 06:28:03